# Advances in the Therapeutic Effects of Apoptotic Bodies on Systemic Diseases

**DOI:** 10.3390/ijms23158202

**Published:** 2022-07-26

**Authors:** Xiaoyan Li, Yitong Liu, Xu Liu, Juan Du, Ujjal Kumar Bhawal, Junji Xu, Lijia Guo, Yi Liu

**Affiliations:** 1Laboratory of Tissue Regeneration and Immunology and Department of Periodontics, Beijing Key Laboratory of Tooth Regeneration and Function Reconstruction, School of Stomatology, Capital Medical University, Beijing 100050, China; xiaoyanli0108@163.com (X.L.); lyt19910731lyt@163.com (Y.L.); liuxu9801@126.com (X.L.); dujuan_1983@163.com (J.D.); uujkl@163.com (J.X.); 2Department of Biochemistry and Molecular Biology, Nihon University School of Dentistry at Matsudo, Chiba 271-8587, Japan; bhawal.ujjal.kumar@nihon-u.ac.jp; 3Department of Pharmacology, Saveetha Dental College, Saveetha Institute of Medical and Technical Sciences, Chennai 600077, India; 4Department of Orthodontics, School of Stomatology, Capital Medical University, Beijing 100006, China; 5Immunology Research Center for Oral and Systematic Health, Beijing Friendship Hospital, Capital Medical University, Beijing 100050, China

**Keywords:** apoptotic bodies, apoptosis, intercellular communication, systemic diseases

## Abstract

Apoptosis plays an important role in development and in the maintenance of homeostasis. Apoptotic bodies (ApoBDs) are specifically generated from apoptotic cells and can contain a large variety of biological molecules, which are of great significance in intercellular communications and the regulation of phagocytes. Emerging evidence in recent years has shown that ApoBDs are essential for maintaining homeostasis, including systemic bone density and immune regulation as well as tissue regeneration. Moreover, studies have revealed the therapeutic effects of ApoBDs on systemic diseases, including cancer, atherosclerosis, diabetes, hepatic fibrosis, and wound healing, which can be used to treat potential targets. This review summarizes current research on the generation, application, and reconstruction of ApoBDs regarding their functions in cellular regulation and on systemic diseases, providing strong evidence and therapeutic strategies for further insights into related diseases.

## 1. Introduction

Apoptosis is a form of programmed process of cell death that occurs throughout life as a part of normal development, which includes distinct cell shrinkage, chromatin condensation, and plasma blebbing [1]. Up to 100–150 billion cells will die each day in the human body via apoptosis to maintain homeostasis [2,3,4,5,6,7,8]. The importance of apoptosis can be inferred from various biological responses and changes, e.g., embryonic development, cell or organ renewal, and turnover. In addition, apoptotic cells may stimulate the proliferation of progenitor cells to improve tissue regeneration and to replace damaged cells [9,10].

Extracellular vesicles (EVs) are recognized as key regulators of intercellular communications and can participate in multiple physiological and pathological processes, such as inflammation, immune responses, coagulation, tumorigenesis, and host–microbe interactions [11,12,13,14,15,16]. Various types of EVs are produced during cell apoptosis, including exosomes, microvesicles, and apoptotic bodies (ApoBDs) [17,18,19,20,21,22]. ApoBDs are specifically generated from apoptotic cells and are capable of encapsulating cellular factors generated during apoptosis [23,24,25]. ApoBDs can be recognized and engulfed by macrophages, dendritic cells (DCs), epithelial cells, endothelial cells, and fibroblasts for clearance purposes, and are subsequently internalized, ingested, and degraded in lysosomes [7,26,27,28], which may facilitate intercellular communications through the transfer of cellular factors [29,30]. In this review, current knowledge about ApoBDs and their therapeutic applications are discussed.

## 2. Cell Apoptosis

All multicellular organisms have a pathway that strictly regulates and controls cell development, maintenance, and clearance. Controlling the number of cells is critically important to maintain the balance between cell proliferation and clearance [31]. Apoptosis is the core of the cell clearance process and plays an important role in normal cell homeostasis. Apoptosis is a physiological form of programmed cell death and is critical for development and tissue homeostasis in animals [1]. Cells actively participate in their own destruction and specific cells are sacrificed for the greater interests of the body. The uniqueness of this gene-directed cell suicide process is to remove residual cell components in adjacent tissues without causing an inflammatory reaction [32,33]. Apoptosis is the key to removing unnecessary cells and to protect cells from significant genotoxic damage.

Apoptosis is mediated by the caspase protein family. Current studies have characterized the intrinsic and extrinsic apoptosis pathways in the body [34] (shown schematically in Figure 1, created with Biorender.com). The intrinsic pathway, also known as the mitochondrial pathway, is mainly involved in development and in genotoxic factor mediated apoptosis, and is regulated by Bcl-2 family members [35,36]. Bcl-2 family members include three subfamilies: pro-apoptotic BH3 only members (Bim, Bid, Puma, Noxa, Hrk, Bmf, and Bad), pro-apoptotic effector molecules (Bax and Bak), and anti-apoptotic Bcl-2 family proteins (Bcl-2, BCL XL, MCL1, A1, and Bcl-b) [37]. Usually, the expression of Bax/Bak is inhibited by anti-apoptotic Bcl-2 family members. After stimulation with apoptosis signals (such as growth and development, lack of growth factors or genotoxic factors), BH3-only members start transcription or post-transcriptional upregulation. Activated BH3-only protein interacts with Bax/Bak or antagonizes anti-apoptotic Bcl-2 family members. Bax/Bak induces the release of cytochrome c from mitochondria. Cytochrome c and APAF-1 form a heptamer complex in a dATP/ATP dependent manner [38,39,40,41], which acts as a scaffold to mediate caspase 9 activation and initiate apoptosis [42,43].

The extrinsic pathway, also known as the death factor pathway, is composed of FasL TNF-α, and TRAIL (TNF-related apoptosis inducing ligand). Cell death signals combine with their specific receptors to produce disc (death inducing signaling complex), which leads to recruitment and activation of caspase 8 and 10. Caspase 9 is activated by the intrinsic pathway and caspase 8 and 10 are activated by the extrinsic pathway to mediate caspase 3 activation. Activated caspase 3 can initiate apoptosis by cleaving more than 1300 cell substrates [44,45,46]. Caspase 7 was considered to be redundant with caspase 3; however, caspase 7 activation requires caspase 1 inflammasomes under inflammatory conditions, while caspase 3 processing proceeds independently of caspase 1 [47].

## 3. The Generation of ApoBDs

Vesicles are one of the earliest morphological changes that can be recognized in the process of apoptosis. The formation of membrane vesicles is the result of an increase of intracellular hydrostatic pressure after actin-mediated contraction [48]. The repeated formation and contraction process of apoptotic cells leads to the production of ApoBDs and fills them with organelles and other cellular contents [49]. The formation of ApoBDs also depends on another phenomenon called AVD (apoptotic volume decrease) [50]. AVD is an early event accompanied by the appearance of membrane vesicles, which leads to the atrophy of apoptotic cells [51,52]. Two stages of AVD have been described; the first stage is characterized by a reversal of the normal Na^+^ and K^+^ gradients, the second stage is related with cytoskeleton organization and further K^+^ extrusion. In this second stage, both cytochrome C release and caspases seem to be involved and required for AVD. AVD is required for cell dismantling into ApoBDs [53]. The inhibition of AVD by destroying the cytoskeleton prevents the formation of ApoBDs [54]. 

Although most apoptotic cells will appear as membrane vesicles, some cells will specifically form other types of membrane protrusions, such as microtubule spikes, beaded apoptotic structures, and so on [48,55]. For example, neurons and some epithelial cells use microtubule spikes instead of membrane vesicles to form ApoBDs. Apoptotic THP-1 cells and primitive human neutrophils show bead-like apoptotic structures. It is worth noting that beaded cell apoptosis seems to be the most effective way to produce ApoBDs, producing about 10–20 ApoBDs at the same time [48]. It has been found that bead-like apoptotic structures and the formation of ApoBDs can be observed in neutrophils that cannot produce membrane vesicles after transformation, which directly proves that the formation of ApoBDs does not require membrane vesicles. However, the appearance of membrane vesicles can promote the disintegration of apoptotic cells and the formation of ApoBDs [56,57,58]. However, the detailed mechanism of the cell division into ApoBDs remains unclear. Researchers believe that the disintegration of apoptotic cells into vesicles requires the combined action of extracellular and intracellular factors as well as some unknown forces to separate membrane processes from the main cell body.

## 4. The Physiological Role of ApoBDs

Phagocytes recognize the “find me” signal sent by apoptotic cells and the “eat me” signal on ApoBDs and phagocytize them [59] to complete the clearance of ApoBDs through ligand/receptor interaction (shown schematically in Figure 2, created with Biorender.com). The process of apoptotic cell clearance is called “efferocytosis”. Efferocytosis can be divided into four stages [60]: the first stage is to locate the target cell. The “find me” signal is a soluble factor released by apoptotic cells that mediates the movement of phagocytes near apoptotic cells. The release of the “find me” signal starts from the beginning of apoptosis and forms a concentration gradient near apoptotic cells [49]. The “find me” signal is recognized by receptors on phagocytes, prompting phagocytes to move to the vicinity of ApoBDs along the concentration gradient. The second stage is the recognition of the “eat me” signal on ApoBDs. The “eat me” signal is comprised of phosphatidylserine (PS) and is externalized. Once the “eat me” signal is recognized and targeted by phagocytes, the phagocytosis process starts [59]. The third stage is phagocytosis. Phagocytes undergo cytoskeletal rearrangements and modifications to enable them to ingest ApoBDs and complete phagocytosis [60]. The last stage is the digestion of cell residues by lysosomal degradation (PS), as an “eat me” signal, can interact with the calcium phospholipid binding protein Annexin V [61]. Annexin V is involved in coagulation because of its extensive binding to PS. Annexin V derivatives have been developed to recognize apoptosis [62,63,64].

Although ApoBDs are easily engulfed by phagocytes, intact apoptotic cells can also be engulfed by phagocytes [65]. Moreover, some cells do not divide to generate ApoBDs. Once the generation of EVs is inhibited, the ability of monocytes and macrophages to clear apoptotic cells will be impaired [65,66].

An increasing amount of evidence has shown that the transfer, circulation, and even the reuse of the contents of ApoBDs widely affects the functions of phagocytes [67]. Like exosomes and microvesicles, ApoBDs also contain residual components of apoptotic cells and play an important role in intercellular communications by transporting signal molecules. ApoBDs can be engulfed by macrophages, DCs, fibroblasts, mesenchymal stem cells (MSCs), endothelium cells, and epithelium cells. Studies have reported that ApoBDs contain DNA, RNA, proteins, cytokines etc. [68,69], which are used for immune activation, recruiting of apoptotic cells, and the regeneration of tissues. After ApoBDs are engulfed, the contents will be released and regulate downstream receptor cells for intercellular communications (shown schematically in Figure 1). 

During apoptosis, the nuclear materials will be distributed into ApoBDs [70]. The horizontal transfer of DNA can occur between different types of adjacent cells, so that DNA materials can be transferred and integrated between cells from different sources to form a new cell genome. For example, DNA contained in lymphoma-derived ApoBDs is engulfed by surrounding fibroblasts, which leads to the integration of lymphoma-derived DNA into the fibroblast genome [68,71]. Many studies have shown that functional molecules (such as DNA, RNA, and protein) can be packaged into ApoBDs and have multiple regulatory functions. Packaging functional molecules into ApoBDs for targeted therapy provides a new approach for future treatments.

## 5. The Therapeutic Effect of ApoBDs on Systemic Diseases

Accumulated reports have revealed the critical role of ApoBDs in intercellular communications by transporting intracellular signaling molecules. The investigation and discovery of ApoBDs has offered new insights for pathological therapeutic targets and explorations for effective treatments (Table 1).

### 5.1. Bacterial Infections in Cancer

Anti-cancer treatments often have a high risk of *Staphylococcus aureus* infections due to surgery, blood transfusions, radiotherapy and chemotherapy, as well as indwelling catheters and drainage tubes [83]. Vancomycin is an efficient treatment for multidrug-resistant *Staphylococcus aureus* (MRSA); however, it can have serious side effects during systemic treatment, including thrombophlebitis, renal injury, and epidermal necrolysis, which can result in more serious poisoning symptoms than the original infection [84]. Loading vancomycin into cancer cell-derived ApoBDs to reconstitute ApoBDs (ReApoBDs) can target cancer cells with the “eat me” signal on ReApoBDs and kill *Staphylococcus aureus* [72].

Studies have shown that cell-derived vesicles (CDVs) extracted from invasive cancer cells can be used for the targeted delivery of miRNAs and nano-contrast agents for treatment and imaging [85,86]. Therefore, a “nano bait” method has been proposed to treat macrophages and tumor cells infected with *Staphylococcus aureus* by using vancomycin loaded ReApoBDs. ReApoBDs were prepared from different cancer cells (SKBR3, MDAMB-231, HepG2, U87-MG and LN229) and were used for vancomycin delivery. In vitro cell culture studies showed that in *Staphylococcus aureus*-infected macrophages, U87-MG and LN229 glioblastoma cell models, ReApoBDs effectively killed intracellular bacteria compared with free vancomycin treatment [72]. Therefore, the recombinant nanocarrier greatly improves the targeting of *Staphylococcus aureus*-infected macrophages and cancer cells. ReApoBDs loaded with vancomycin have the potential to kill intracellular *Staphylococcus aureus* infections and that strategy can be used to eliminate treatment-related MRSA infection during anti-cancer treatment.

### 5.2. Atherosclerosis

The development of atherosclerosis is mainly due to the accumulation of lipids and inflammatory debris in vascular walls; however, it is also related to the apoptosis of macrophages, smooth muscle cells (SMCs), and endothelial cells. Studies have shown that ApoBDs generated by apoptotic endothelial cells possess the ability to regulate atherosclerosis [73].

Pro-inflammatory cytokine IL-1α was found in endothelial cell-derived ApoBDs, and moreover, the ApoBDs can induce monocytes to secrete chemokines, as well as to mediate neutrophil inflammation [87]. Although the presence of ApoBDs will further exacerbate inflammation and accelerate atherosclerosis, they may also work as mediators for vascular repair through the recruitment of endothelial progenitor cells [74]. Studies have shown that ApoBDs can promote the differentiation of endothelial progenitor cells. ApoBDs can transfer miRNA-126 to recipient vascular cells, promote the production of CXCL12 and recruit progenitor cells for tissue repair [73]. However, due to the lack of an accurate, repeatable, and standardized technology for the isolation of ApoBDs, the application of ApoBDs in atherosclerotic diseases needs to be further investigated.

### 5.3. Bone Homeostasis

During the bone remodeling process, the lifespan of osteoclasts is about 2 weeks after which they undergo apoptosis [88,89], which will generate a large number of ApoBDs. It has been shown that bone marrow monocytes (BMMs) are stimulated by nuclear factor kappa B receptor activator ligand (RANKL) to differentiate into pre-osteoclasts (pOCs) and mature osteoclasts (mOCs). Sodium alendronate (AlN) has been used to induce the apoptosis of pOCs and mOCs and to generate their ApoBDs, respectively. It was found that ApoBDs derived from mOCs can be engulfed by MC3T3-E1 pre-osteogenic cells and promotes their viability. Among all EVs derived from osteoclasts, mOCs-ApoBDs have a high concentration of RANK and possess the highest osteogenic potential. Mechanistically, it was also revealed that mOC-ApoBDs induce osteoblast differentiation via the PI3K/Akt/mTOR/S6K signaling pathway, which confirmed the osteogenic promoting ability of mOC-ApoBDs [75].

Hence, ApoBDs are necessary to maintain systemic bone density. Studies of Fas-deficient MRL/lpr and caspase 3^−/−^ mice revealed that a reduced generation of ApoBDs significantly impairs the self-renewal and osteogenic/adipogenic differentiation of bone marrow MSCs. The systemic infusion of exogenous ApoBDs rescued the impairment of MSCs and ameliorated the osteopenia phenotype. MSCs were able to engulf apoptotic bodies via integrin αvβ3 and reuse ApoBD-derived ubiquitin ligase RNF146 and miR-328-3p to inhibit Axin1. MSCs thereby activate the Wnt/β-catenin pathway. This suggests the potential use of ApoBDs in the treatment of osteoporosis [76].

### 5.4. Hepatic Fibrosis

Hepatic stellate cells (HSCs) play a key role in the process of liver fibrosis. Under the induction of fibrogenic stimulation, HSCs were activated and differentiated into myofibroblasts. HSC survival is central to the progression of liver fibrosis and the induction of HSC apoptosis can rescue liver fibrosis. Phagocytosis of ApoBDs promotes HSC survival via the JAK1/STAT3 and PI3K/Akt/NF-κB pathways [77]. Targeted treatment can contribute to the recovery of liver fibrosis.

### 5.5. Enhancement of the Effect of Chemotherapy Drugs

Chemotherapeutic nanomedicines can exploit the neighboring effect to increase tumor penetration; however, the neighboring effect is limited due to the consumption of chemotherapeutic drugs and the drug resistance of internal hypoxic tumor cells. ApoBDs were shown to carry the remaining drugs into neighboring tumor cells after apoptosis [78]. It was reported that camptothecin (CPT) could kill tumor cells with a normal external oxygen content to produce ApoBDs, while the hypoxia-activated prodrug PR104A remained active. The remaining drugs can be effectively transported into tumor cells through ApoBDs. This ApoBD-mediated neighboring effect provides a new approach to enhance the efficacy of chemotherapeutic drugs, which may improve the therapeutic efficiency of clinical nano drugs in the future.

### 5.6. Immunotherapy and Immune Defense

In pathological conditions such as cancers and infections, ApoBDs contain antigens from cancer cells and pathogens, which can be recognized by the immune system. Since ApoBDs are easily engulfed by antigen-presenting phagocytes such as DCs, they can activate an adaptive T cell response through the cross antigen-presenting process [90,91]. Macrophages will undergo apoptosis after a mycobacteria infection. The ApoBDs generated from apoptotic macrophages contain pathogen antigens, which can trigger dendritic cell-mediated cross presentation and CD8+ T cell activation through MHC-I and CD11b [79]. Infections can be controlled by inhibiting the apoptosis of macrophages, which not only confirms that ApoBDs possess a strong ability to initiate adaptive immune responses, but also reveals their potential application in vaccine development and immunotherapy.

Meningeal epithelial cells (MECs) are the cellular components of the meninges, which work as a barrier for the central nervous system, establish an interface between neuronal tissues and cerebrospinal fluid, and are also a part of the immune system. MECs are highly activated phagocytes, which can engulf and digest apoptotic cells. MECs are immune suppressive via the inhibited secretion of pro-inflammatory, chemoattractant cytokines and chemokines following the uptake of apoptotic bodies to shut down immune responses in the brain [80].

### 5.7. Diabetes

Type I diabetes is due to the autoimmune destruction of insulin-producing B cells in the islets. An ideal immunotherapy should inhibit the autoimmune destruction, avoid systemic side effects, and promote islet regeneration. Apoptotic cells, as the source of autoantigens, are cleared rapidly by macrophages and DCs through an immunologically silent process that contributes to maintaining tolerance. DCs that produce peripheral immune tolerance may open new therapeutic approaches to prevent or alleviate autoimmunity.

In one study, immature DCs were obtained from the bone marrow of non-obese diabetes (NOD) mice and were pulsed with antigen-specific ApoBDs from the beta cell line NIT-1 [81]. The DCs with phagocytosed apoptotic cells reduced the expression of co-stimulatory molecules CD40 and CD86 and proinflammatory cytokines, failed to complete the antigen presentation process, rebuilt peripheral immune tolerance, and reduced the incidence rate of diabetes in NOD mice. Those findings proved the regulatory role of ApoBDs in type I diabetes and opened new therapeutic approaches for the prevention or remission of autoimmunity.

### 5.8. Wound Healing

To identify new treatments for large-area skin wounds, an increasing number of studies has shown that bone marrow MSCs can accelerate the healing of skin wounds and restore an intact and orderly skin structure [92]. ApoBDs derived from bone marrow MSCs further enhanced the migration and proliferation of fibroblasts and accelerated skin wound healing by inducing the polarization of macrophages to the M2 phenotype [82].

## 6. Engineering and Recombination of ApoBDs

Engineered and recombinant EVs carrying effective therapeutic molecules are ideal candidates for the treatment of diseases (shown schematically in Figure 3, created with Biorender.com). They can amplify the targeting and efficacy of EVs treatments, as well as maximize the therapeutic roles of EVs to disrupt the process of disease development.

Several studies have reported that the recombination of EVs can be achieved by modifying the donor cells [93,94,95,96,97,98,99,100]. In addition, to engineer the construction of EVs, multiple methods for therapeutic molecule loading can be used. Catalase-loading is one construction method for EVs that has proven to be a versatile strategy to treat inflammatory and neurodegenerative disorders [101]. Electroporation can be used to load target exogenous siRNAs into EVs, and intravenously injected engineered EVs can result in the knockdown of specific genes and a therapeutic potential of recombinant EV-mediated siRNA delivery has been observed [102]. Sonication has been used as an approach to actively load functional small RNAs into EVs with minimal detectable aggregation, which can then be taken up by recipient cells and are capable of targeting mRNA knockdown leading to reduced protein expression [103]. Chimeric ApoBDs functionalized with a natural membrane and modular delivery system can be applied for the modulation of inflammation [104]. The combination of natural neutrophils and mesoporous silica nanoparticles loaded with hexyl 5-aminolevulinate hydrochloride (HAL) exhibited excellent inflammation-tropism and immunoregulatory properties in myocardial infarction [105].

However, the existing methods still have considerable limitations: (a) residual contents irrelevant to the purpose of treatment cannot be completely removed, which may pose a potential threat to the body [106]; (b) EVs activate downstream signals to regulate receptor cells only after the receptor cells release the bioactive molecules contained therein. However, EVs entering receptor cells can be intercepted by the lysosomal system, which reduces the content of EVs in the cells and significantly reduces their therapeutic effect [107]; and (c) surface modified chemical bonds that increase the targeting of EVs may lead to the destruction of the membrane structure [108].

## 7. Concluding Remarks and Prospects

Extensive research studies on EVs have proven that ApoBDs not only phagocytize residual materials of apoptotic cells, but, like exosomes and microvesicles, ApoBDs also play an important role in intercellular communications by transporting signaling molecules. The natural combination of ApoBDs and phagocytes provides a unique opportunity for future treatment schemes. They are bound to provide new ideas for immunotherapy, vaccine development, tissue regeneration, drug delivery, and disease diagnosis, which can maximize the therapeutic effects and specificity.

Although the applications of ApoBDs have great prospects, there are still some key points that need to be further investigated, including their heterogeneity, storage conditions, quality control, standardized isolation, and purification. It is necessary to further understand the biological mechanism of ApoBDs. Various molecules have been proposed as signals of phagocytosis; however, the consensus remains unclear. Further investigation of the targeting, classification, and engulfing mechanism of ApoBDs will magnify their therapeutic effects and overcome those limitations. ApoBDs play regulatory roles in multiple physiological and pathological processes, and the manipulation and reconstruction of ApoBDs hold great promise for curing systemic diseases.

## Figures and Tables

**Figure 1 ijms-23-08202-f001:**
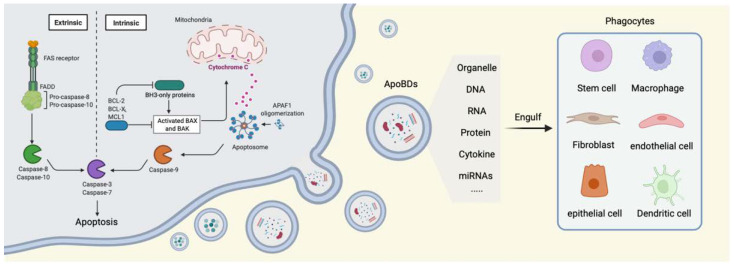
ApoBDs derived from apoptotic cells induced by extrinsic and intrinsic pathways participating in intercellular communications.

**Figure 2 ijms-23-08202-f002:**
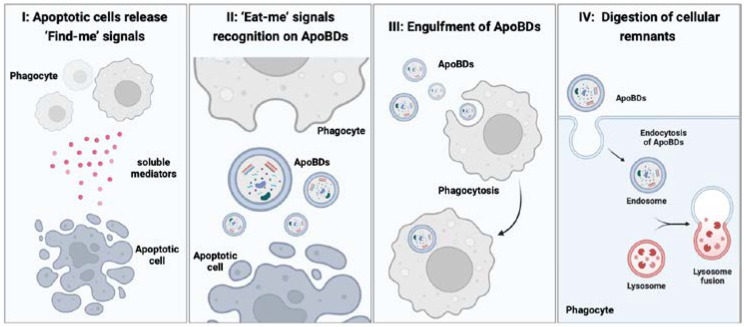
Four stages of efferocytosis. Stage I: Apoptotic cells release soluble mediators to attract phagocytes; Stage II: The “Eat-me” signals present on ApoBDs surface for recognition and anchorage of phagocytes; Stage III: Cytoskeletal rearrangement and modification of the phagocytes occur to enable ingestion of ApoBDs; Stage IV: Digestion of cellular remnants through lysosomal degradation.

**Figure 3 ijms-23-08202-f003:**
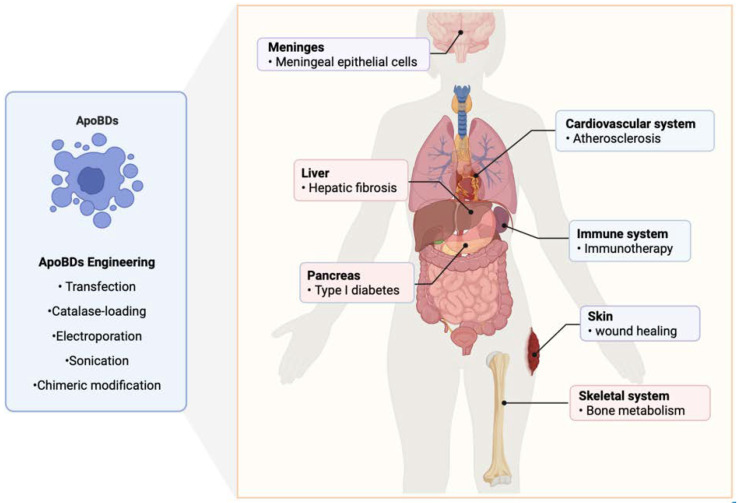
ApoBDs are implicated in systematic diseases. ApoBDs are involved in the physiological and pathological processes and the application of engineering ApoBDs possessed the therapeutic effect of systematic diseases.

**Table 1 ijms-23-08202-t001:** The regulatory mechanism of ApoBDs on systematic diseases.

Systematic Condition	ApoBDs	Regulatory Mechanism	Reference
Bacterial infection	Vancomycin loading cancer cell-derived ApoBDs	Targets the “eat me” signal on ApoBDs and vancomycin is delivered to kill Staphylococcus aureus	[72]
Atherosclerosis	Endothelial cell-derived ApoBDs	Promotes endothelial progenitor cell proliferation and differentiation	[73,74]
Bone homeostasis	1. mOC-derived ApoBDs2. Circulating ApoBDs	1. Induces osteoblast differentiation via the PI3K/Akt/mTOR/S6K signaling pathway2. Circulating ApoBDs maintain the self-renewal and osteogenic/adipogenic differentiation of BMMSCs via the Wnt/β-catenin pathway	[75,76]
Hepatic fibrosis	HepG2-derived ApoBDs	Promotes HSC survival via the JAK1/STAT3 and PI3K/Akt/NF-κB pathways	[77]
Chemotherapy	CPT+ PR104A loading cancer cell-derived ApoBDs	Enhances the ApoBD-based neighboring effect and facilitates the deep penetration of chemotherapeutic agents	[78]
Immunotherapy	1. Macrophage-derived ApoBDs2. U-937/SH-SY5Y-derived ApoBDs	1. Triggers dendritic cell-mediated cross presentation and CD8 + T cell activation through MHC-I and CD11b2. Inhibits the secretion of pro-inflammatory, chemoattractant cytokines and chemokines	[79,80]
Diabetes	NIT-1-derived ApoBDs	Reduces the expression of co-stimulatory molecules CD40, CD86, and proinflammatory cytokines of DCs, rebuilds peripheral immune tolerance	[81]
Wound healing	BMMSC-derived ApoBDs	Enhances the migration and proliferation of fibroblasts, inducing the polarization of macrophages to the M2 phenotype	[82]

ApoBDs: apoptotic bodies; mOC: mature osteoclast; BMMSC: bone marrow mesenchymal stem cell.

## Data Availability

The datasets used and analyzed in this study are available from the corresponding authors (lililiuyi@163.com) on reasonable request.

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
