# Peer review of "Advances in the Therapeutic Effects of Apoptotic Bodies on Systemic Diseases"

_ijms, 2022, doi:10.3390/ijms23158202_

Round 1
Reviewer 1 Report
Manuscript is well written and clear. After minor revision I can recommend it for publication.
1) Are there any studies suggesting that not only Bcl-2 family can activate appoptosis?
2) Based on what signals cell goes thorugh intrinsic or extrinsic pathway?
3) Does both intrinsic and extrinsic pathway leads to the same ApoBDs or there are some differences?
4) In the text for Figure 1 authors did not descibe role of Caspase-7 and -10.
5) Please add in what programe figures were created
6) Authors described that ApoBDs can have different size and shape. How/ does it affect their signal transduction/ interaction
7) Please describe AVD process in more detail. How the process is iniciated? Does caspases play role in that?
8) Paragraph 4 of the article focus on the ApoBDs phisiological role. Please add figure that describe the process of "efferocytosis".
9) How signals "find me" and "eat me" are send? Though receptor/ ligand? Please add more detail
10) How ApoBDs can used as a vehicles/vectors? What would be their capacity?
Author Response
Manuscript is well written and clear. After minor revision I can recommend it for publication.
1) Are there any studies suggesting that not only Bcl-2 family can activate apoptosis?
RESPONSE: We appreciate this suggestion. Cell apoptosis was regulated by intrinsic (Bcl-2 family members) and extrinsic (FasL, TNF-α, and TRAIL) pathways. Besides the classic pathways, studies reported that the dysregulation of survival pathway PI3K/Akt/mTOR was frequently observed in tumors. Pro-apoptotic regulators such as p53 and BIM were also possessed regulatory role in cancer cell apoptosis.
2) Based on what signals cell goes through intrinsic or extrinsic pathway?
RESPONSE: The authors appreciate the Reviewer for pointing out these oversights. The intrinsic pathway associates with the mitochondrial outer membrane and the endoplasmic reticulum/nuclear membrane. Intrinsic pathway proteins are sentinel that detect developmental death cues or intracellular damage [1]. Whereas the extrinsic pathway is discovered to be an important aspect of the immune response, and family members such as FASL and TRAIL regulate activation-induced apoptosis of immune system [2].
Reference:
- Cory S, Adams JM. The Bcl2 family: regulators of the cellular life-or-death switch. Nat Rev Cancer. 2002 Sep;2(9):647-56.
- Ashkenazi A. Targeting death and decoy receptors of the tumour-necrosis factor superfamily. Nat Rev Cancer. 2002 Jun;2(6):420-30.
3) Does both intrinsic and extrinsic pathway leads to the same ApoBDs or there are some differences?
RESPONSE: The authors recognize the Reviewer’s concern. Strong evidence showed that the donor cell type and the way of donor cell apoptosis have great impact on the size, composition, and function of ApoBDs [3,4].
Reference:
- Gregory CD, Dransfield I. Apoptotic Tumor Cell-Derived Extracellular Vesicles as Important Regulators of the Onco-Regenerative Niche. Front Immunol. 2018 May 23;9:1111.
- O'Brien K, Breyne K, Ughetto S, Laurent LC, Breakefield XO. RNA delivery by extracellular vesicles in mammalian cells and its applications. Nat Rev Mol Cell Biol. 2020 Oct;21(10):585-606.
4) In the text for Figure 1 authors did not describe role of Caspase-7 and -10.
RESPONSE: We appreciate the Reviewer’s suggestions. The role of Caspase 7 and 10 has been described in the revised manuscript (Page 2, Paragraph 3) and highlighted with yellow color. We hope that you find these revisions an improvement.
5) Please add in what program figures were created
RESPONSE: We appreciate this suggestion. The figures were created in the https://app.biorender.com and was cited in the revised manuscript.
6) Authors described that ApoBDs can have different size and shape. How does it affect their signal transduction/ interaction
RESPONSE: Thank you for your thorough review and salient observations. ApoBDs play a critical role in intercellular communications by transporting intracellular signaling molecules, this process mainly dependent on the special molecules display on ApoBDs surface and the inside components. The ApoBDs derived from different donor cells exhibits different size, shape, surface molecule and cargo, which will transport different signal during intercellular communications.
7) Please describe AVD process in more detail. How the process is iniciated? Does caspases play role in that?
RESPONSE: The authors appreciate the Reviewer for pointing out this oversight. We described AVD process in detail in the revised manuscript (Page 3, Paragraph 1) and highlighted with yellow color. We hope that you find these revisions an improvement.
8) Paragraph 4 of the article focus on the ApoBDs physiological role. Please add figure that describe the process of "efferocytosis".
RESPONSE: We appreciate this suggestion. The figure was added in the manuscript as “Figure 2”. The comment has led to a stronger and clearer revised manuscript.
9) How signals "find me" and "eat me" are send? Though receptor/ ligand? Please add more detail
RESPONSE: We greatly appreciate these helpful comments. Accumulating studies have identified several “find me” and “eat me” signals during efferocytosis including the lipid lysophosphatidylcholine (LPC) and its receptor G2A [5]; sphingosine 1-phosphate (S1P) and its receptor S1P1 [6]; CX3CL1 and its receptor CX3CR1 [7]; lysoPC and its receptor G2A [8] and the nucleotides ATP and UTP [9].
Reference:
- Witte ON, Kabarowski JH, Xu Y, Le LQ, Zhu K. Retraction. Science. 2005 Jan 14;307(5707):206.
- Weigert A, Cremer S, Schmidt MV, von Knethen A, Angioni C, Geisslinger G, Brüne B. Cleavage of sphingosine kinase 2 by caspase-1 provokes its release from apoptotic cells. Blood. 2010 Apr 29;115(17):3531-40.
- Truman LA, Ford CA, Pasikowska M, Pound JD, Wilkinson SJ, Dumitriu IE, Melville L, Melrose LA, Ogden CA, Nibbs R, Graham G, Combadiere C, Gregory CD. CX3CL1/fractalkine is released from apoptotic lymphocytes to stimulate macrophage chemotaxis. Blood. 2008 Dec 15;112(13):5026-36.
- Peter C, Waibel M, Radu CG, Yang LV, Witte ON, Schulze-Osthoff K, Wesselborg S, Lauber K. Migration to apoptotic "find-me" signals is mediated via the phagocyte receptor G2A.
- Elliott MR, Chekeni FB, Trampont PC, Lazarowski ER, Kadl A, Walk SF, Park D, Woodson RI, Ostankovich M, Sharma P, Lysiak JJ, Harden TK, Leitinger N, Ravichandran KS. Nucleotides released by apoptotic cells act as a find-me signal to promote phagocytic clearance. Nature. 2009 Sep 10;461(7261):282-6.
10) How ApoBDs can used as a vehicles/vectors? What would be their capacity?
RESPONSE: The authors recognize the Reviewer’s concern. Numerous studies reported that ApoBDs carry a variety of cargo, including RNAs, proteins, lipids, and DNA, which can be taken up by other cells, both in the direct vicinity of the source cell and at distant sites in the body via biofluids, and elicit a variety of phenotypic responses. Owing to their unique biology and roles in cell-cell communication, ApoBDs have attracted strong interest, which is further enhanced by their potential clinical utility.
Thank you for your detailed comments and suggestions. We found them quite useful as we revised out manuscript. We are grateful for the time and energy you expended on our behalf.

Reviewer 2 Report
The review by Li et al. aimed at discussing the role and therapeutic applications of apoptotic bodies. It is quite a well-written manuscript that can be interesting for the scientific community. However, the title is misleading; it suggests a comprehensive review. A title starting with “advances in” or “highlights of” would be more appropriate. There are some additional minor points:
Line 44: References 17 and 21 are not adequate.
Line 63: Correct “that” to “the”.
Lines 148-150: Cite the original report rather than a review (reference 70).
Author Response
The review by Li et al. aimed at discussing the role and therapeutic applications of apoptotic bodies. It is quite a well-written manuscript that can be interesting for the scientific community. However, the title is misleading; it suggests a comprehensive review. A title starting with “advances in” or “highlights of” would be more appropriate. There are some additional minor points:
Line 44: References 17 and 21 are not adequate.
Line 63: Correct “that” to “the”.
Lines 148-150: Cite the original report rather than a review (reference 70).
RESPONSE: We thank you for your excellent comments and suggestions, which have led to a stronger and clearer revised manuscript. We revised the manuscript according to all these suggestions and highlighted with yellow color.
We appreciate your insightful comments. We worked hard to respond to all of them. Thank you for taking the time and energy to help us improve our paper.
